# Effectiveness of antiresorptive medications in women on long-term dialysis after hip fracture: A population-based cohort study

Yu-Ciou Lin[1], Tien-Ching Lee[2,3,4,5], Chung-Yu Chen[6], Shun-Jin Lin[1,6], Shang-Jyh Hwang[3,7,8,9], Ming-Yen Lin[7,8,10]*

1 Graduate Institute of Clinical Pharmacy, College of Pharmacy, Kaohsiung Medical University, Kaohsiung, Taiwan, 2 Orthopaedic Research Center, College of Medicine, Kaohsiung Medical University Hospital, Kaohsiung Medical University, Kaohsiung, Taiwan, 3 Graduate Institute of Medicine, College of Medicine, Kaohsiung Medical University, Kaohsiung, Taiwan, 4 Department of Orthopedics, Kaohsiung Municipal Ta-Tung Hospital, Kaohsiung Medical University, Kaohsiung, Taiwan, 5 Department of Orthopedics, Kaohsiung Medical University Hospital, Kaohsiung Medical University, Kaohsiung, Taiwan, 6 School of Pharmacy, College of Pharmacy, Kaohsiung Medical University, Kaohsiung, Taiwan, 7 Division of Nephrology, Department of Internal Medicine, Kaohsiung Medical University Hospital, Kaohsiung Medical University, Kaohsiung, Taiwan, 8 Department of Renal Care, College of Medicine, Kaohsiung Medical University, Kaohsiung, Taiwan, 9 Institute of Population Health Sciences, National Health Research Institutes, Miaoli, Taiwan, 10 Master of Public Health Degree Program, College of Public Health, National Taiwan University, Taipei, Taiwan

* mingyenlin3@gmail.com

**Data Availability Statement:** These Data used in our study were obtained from the National Health Insurance Research Database (NHIRD), maintained by Taiwan National Health Research Insurance

## Abstract

There is no clear evidence how effective the antiresorptive (AR) drugs alendronate and raloxifene are at reducing risk of second hip fracture and mortality in dialysis populations. The purpose of this study was to compare the risk of hospitalization for second hip fracture and risk of mortality between AR user and non-user groups in Taiwanese women on long-term dialysis with hip fractures. We conducted a retrospective cohort study using Taiwan National Health Insurance Research Datasets. Long-term dialysis women older than 50 years with newly diagnosed hip fractures and new to AR therapy from 2005 to 2011 were recruited. The patients were divided into AR users and non-users and matched by propensity score. We used Cox Proportional Hazards models to assess association of AR with risks of second hip fracture and mortality. Totally, 1,079 dialysis patients were included, and after matching, we were left with 74 AR users and 74 non-users. AR users did not show a significant reduction in the incidence of second hip fracture compared with non-users (adjusted Hazard Ratio (HR): 0.91, 95% CI: 0.30–2.76), and alendronate users exhibited higher risk of second hip fracture compared with raloxifene users (adjusted HR: 2.80, 95% CI: 0.42–18.79). In addition, AR users were found to have significantly lower 1- and 2-year mortality rates than the non-users (1- year: adjusted HR 0.25, 95% CI, 0.07–0.90; 2-year: 0.35, 95%CI: 0.17–0.72). AR treatment did not significantly improve the risk of second hip fracture but significantly reduce mortality in older women on dialysis. Further clinical trials on effectiveness of AR medications for dialysis populations should be warranted.

(NHRI). The NHRI has stopped providing this service and transferred these data to the Health and Welfare Data Science Center (HWDC) since June 28th, 2016. Due to legal restrictions imposed by the government of Taiwan in relation to the "Personal Information Protection Act", these data cannot be made publicly available. Data can be acquired through formal application to the HWDC, Department of Statistics, Ministry of Health and Welfare, Taiwan (https://dep.mohw.gov.tw/DOS/np-2497-113.html).

**Funding:** Any grants or fellowships supporting the writing of the paper: We would like to thank the Kaohsiung Medical University Hospital, Kaohsiung Medical University, Taiwan National Health Research Institutes, and Taiwan Ministry of Science and Technology for partly supporting this study. SJH is supported by Kaohsiung Medical University Hospital, Kaohsiung Medical University (grant number: KMUH104-4R09 and KMUH105-5R16), Taiwan National Health Research Institutes (grant number: NHRI-EX108-10505PI) and Taiwan Ministry of Science and Technology (grant number: MOST: 104-2314-B-037-073- MY3 and MOST: 107-2314-B-037-020-MY2). MYL is supported by Kaohsiung Medical University Hospital, Kaohsiung Medical University (grant number: KMUH105-5R17 and KMUH106-6R21), Taiwan Ministry of Science and Technology (grant number: MOST:105- 2314-B-037-065-, MOST: 106-2314-B-002-253-, MOST:107-2314-B-037-122- and MOST- 108-2314-B-037-110). The funders had no role in the study design, data collection and analysis, decision to publish, or preparation of the manuscript.

**Competing interests:** The authors have declared that no competing interests exist.

## Introduction

Osteoporosis is a systematic disease that causes decreased bone strength and risk of fractures [1]. According to a previous report, one in three women and one in five men have at least one vertebral, hip, or wrist fracture in their lifetimes [2]. Risk of mortality following hip fracture is estimated to be at least double for age-matched control populations [3], with the main cause of death being infection due to being bedridden over a long period. Identifying high risk groups and effective treatment methods is vital for fracture prevention and mortality reduction.

Beside gender and age, renal function impairment has been mentioned as being highly related to excessive risk of hip fracture [4]. It has been estimated that globally, over 850 million people live with renal function impairment, and within this patient group, over 3.9 million require renal replacement therapies [5]. Although high risk of refracture with quite poor prognosis is anticipated in this population, recent clinical trials usually list patients with low renal function as exclusion criteria, resulting in little scientific evidence to suggest appropriate clinical practice; therefore, results from real-world data offer opportunities to bridge this gap and add primary information of effective size for future randomized clinical trial design.

Antiresorptive (AR) medications (e.g., alendronate or raloxifene) have been well-recognized to improve surrogate indicators of fracture (e.g., bone mineral density) [6–10] and prevent vertebral fracture and nonvertebral fracture in the general population [11, 12]. However, little is known about its effects in patients whose estimated glomerular filtration rate is less than 15 ml/min/1.73 m$^2$. We conducted a population-based study to evaluate the effect of AR on the risk of second hip fracture and the risk of mortality in older Taiwanese women on dialysis.

## Materials and methods

### Data source

The study used data from a longitudinal health insurance database, Taiwan's National Health Insurance Research Database, which contains the healthcare claim records of more than 99% of the residents in Taiwan. The Catastrophic Illness Registry is a national registry system to waive copayment for patients diagnosed with "catastrophic illnesses", e.g., cancer, end-stage renal disease, and autoimmune diseases. We extracted data for all cases diagnosed with end-stage renal disease from the Registry of Catastrophic Illness Database and kept those patients who had the prescription of dialysis treatment for more than 90 days to ensure that only patients with long-term dialysis were included.

### Ethics statement

This study was approved by the Institutional Review Board of the Kaohsiung Medical University Hospital (KMUHIRB-EXEMPT (II)-20170015). All patient data were de-identified, so the need for informed consent was waived.

### Study design and population

All female patients with age above 50 years and having long-term dialysis from 2005 to 2011 were identified. Among these patients, only patients who had been hospitalized for newly diagnosed hip fracture (ICD-9 codes 820) and receiving primary hip fracture surgery involving hemiarthroplasty (ICD-9-CM Procedure Code 81.52) or internal fixation (ICD-9-CM Procedure Code 79.15, 79.35) during hospitalization were enrolled in this study. The index date was defined as the date of hospitalization for newly diagnosed hip fracture. We excluded all patients who had received renal transplantation before dialysis as well as all patients who had

circumstances that might influence the hip fracture evaluation and severity, including fractures resulting from car accidents or high-impact trauma (ICD-9 codes E810-E819, E881-E884), Paget's disease (ICD-9 codes 731.0), or malignant neoplasm (ICD-9 codes 140–208). We further excluded those with hip fractures prior to dialysis, patients not found to use the drugs persistently, as well as those exposed to other osteoporosis medications (ibandronic acid, zoledronic acid, teriparatide, calcitonin, clodronate, pamidronate, risedronate).

## Exposure assessment

We established a one-year washout period duration for AR prescription. Patients who were newly prescribed AR (alendronate and raloxifene) within 90 days of the index date and had at least three-monthly claims records for AR prescriptions within 30 days after the first prescription date were defined as AR users. Our primary analysis used the on-treatment scenario; that is, patients were censored if they switched to other osteoporosis medications once AR treatment had begun. Patients who did not receive osteoporosis medications during the study period were defined as non-users. Total supply in days and quantity of drugs for both inpatients and outpatients were estimated based on pharmacy claims.

## Covariates

We collected demographic information obtained at the beginning of treatment and information of other covariate variables from medical and pharmacy claims during the year leading up to the index date. Covariates were demographic characteristics (age, duration of dialysis, socioeconomic status), osteoporosis, fracture history (nonvertebral fractures other than radius/ulna or hip fracture), comorbidities that might influence fracture risk (peripheral neuropathy, Parkinson's disease, cardiovascular disease, cerebrovascular disease, diabetes mellitus, hypertension, chronic pulmonary disease, cataracts, dementia, mental disorders, rheumatic arthritis, chronic liver disease), and co-medications (glucocorticoids, antiepileptic drugs, anti-depressants, beta-blocking agents, benzodiazepine, analgesics, sedatives and hypnotics, statins, vitamin D and analogues, diuretics, oral diabetes medications, proton pump inhibitors, nitrates, oral anticoagulants, oral non-steroidal anti-inflammatory drug). Socioeconomic status was defined by insurance amount divided into New Taiwan Dollars (NT$) ≤15,840, 15,841–25,000, and ≥25,001 per month respectively [13, 14]. Osteoporosis, fracture history, and comorbidity were defined based on ICD-9-CM codes (S1 Table) listed on ≥ 2 ambulatory care claims records or ≥ 1 inpatient care claims records during the year prior to the index date. The listed co-medicines were identified based on Anatomical Therapeutic and Chemical codes (S2 Table). Patients were co-medication users when listed medicines prescribed over 28 defined daily doses (DDD).

## Outcomes

Our outcome of interest was the risk of hospitalization for second hip fracture and risk of mortality. Second hip fracture was identified by diagnosis code and procedure codes 79.15, 79.35, and 81.25 on claims. Mortality was defined as death during hospitalization or patients discharged in critical conditions and disenrolled in NHI within 3 days after discharge. Study participants were followed from the index date until diagnosis of second hip fracture, death, or final prescription claim before the end of 2012, whichever happened first.

## Statistical analysis

Descriptive statistics including mean (standard deviation) and count (percentage) were used to describe distributions of patient characteristics between the AR user and non-user groups

and tested the significant differences by independent *t*-test and Chi–square test respectively, while the Kaplan-Meier method was used to estimate cumulative incidence of second hip fracture and mortality. A series of univariable and multivariable Cox proportional hazards models analyses were used to compare the risks of second hip fracture and mortality among AR users compared to AR non-users and raloxifene users compared to alendronate users. First of all, we used multivariable regression models to adjust for significant covariates with criterion P value <0.1 in univariable Cox-regression (S3 Table). Then, propensity score (PS) of AR use was calculated by considering all covariates listed in Table 1 as independent variables through multiple logistic regression with forced entry approach. To increase statistical power, PS matching pair from 1:1 to 1:5 using the Greedy 5 to 1 digit technique [15] was taken into consideration. Finally, 1:1 matching pair with similar propensity scores was chosen to maintain the largest case number of the AR user group. Finally, we used competing risk approach adjustment by cumulative incidence function to analyze the risk of second hip fracture [16, 17]. Proportional hazard assumptions were inspected using interaction terms between exposures and time.

Because patients who switched to other osteoporosis medications were considered censored, the covariates were adopted as time-dependent factors for adjusting different follow-up periods. In our calculations of the associations among drug adherence, persistence, risk of second hip fracture and risk of mortality, proportion of days covered (PDC) was assumed to be a measure of medication adherence [18]. The PDC was calculated by the sum of unique days with AR supply divided by the total number of days in the observed period. The persistence was defined by AR prescription refilled within 30 days between the last prescription date of AR and the next prescription [19, 20]. We stratified AR users by their times of prescription refills (≥3, ≥6, and ≥9 times) to estimate changes of main results.

Several sensitivity analyses were performed to evaluate the robustness of our findings. Intention to treat was analyzed based on the assumption that patients' exposure to medications continued to the end of follow-up. Duration of therapy was extended to the last date covered by drug use plus 90 days. The patients who switched to other osteoporosis medications were considered censored in the primary analysis, and these patients were further excluded in sensitivity analysis to confirm the exact drug effect. Patients included at index date had to survive (be alive or event-free) until criteria defining treatment was met, which might create some possible immortal time bias, or until the time span from prescription discontinuation to study outcomes occurred, which might involve some lag time bias; therefore, Cox regression was used to adjust for possible immortal time and lag time bias in the analysis [21]. Finally, some potential risk factors for fractures were easily measured, so fracture risk was also compared separately by age group (50–65, 65–80, ≥80 years), socioeconomic status (<NT$ 15,840, ≥ NT$15,840), fracture history (yes, no), co-glucocorticoids use (yes, no) and unbalanced variables after propensity score matching. All statistical operations were performed using SAS 9.4. (SAS Institute Cary, North Carolina). A 2-side p-value less than 0.05 was considered as statistically significant.

## Results

### Study cohort and patient characteristics

This study identified 112,883 patients on long-term dialysis during the study period from 2004 to 2011, and 7,463 patients with newly diagnosed hip fractures were also found during this period. Male patients (N = 2,121) and patients with Paget's disease (N = 1) were excluded, as were those with fractures due to car accidents or high impact trauma (N = 42), malignant neoplasm (N = 1,813), and having no dialysis records after hip fracture (N = 467), leaving us with 2,992 of the women on long-term dialysis with hip fracture. After excluding those with hip

**Table 1. Baseline characteristics of AR users and non-users.**

| | Full cohort | | | Matched cohort | | |
|---|---|---|---|---|---|---|
| | AR non-users (n = 1005) | AR users (n = 74) | P-value | AR non-users (n = 74) | AR users (n = 74) | P-value |
| Age, year, mean (SD) | 72.78 (8.84) | 72.80 (9.94) | 0.99 | 73.37 (8.26) | 72.80 (9.94) | 0.71 |
| 50–64, n (%) | 183 (18.21) | 15 (20.27) | 0.90 | 12 (6.22) | 15 (20.27) | 0.69 |
| 65–79, n (%) | 578 (57.51) | 41 (55.41) | | 46 (62.16) | 41 (55.41) | |
| >80, n (%) | 244 (24.28) | 18 (24.32) | | 16 (21.62) | 18 (24.32) | |
| Duration of dialysis, mean (SD) | 2.90 (2.10) | 2.70 (1.92) | 0.42 | 2.83 (2.12) | 2.70 (1.92) | 0.69 |
| Socioeconomic, NT$, n (%) | | | 0.23 | | | 0.35 |
| ≤15, 840 | 676 (67.26) | 43 (58.11) | | 50 (67.57) | 43 (58.11) | |
| 15, 841–25, 000 | 312 (31.04) | 30 (40.54) | | 22 (29.73) | 30 (40.54) | |
| ≥ 25, 001 | 17 (1.69) | 1 (1.35) | | 2 (2.70) | 1 (1.35) | |
| Comorbidities, n(%) | | | | | | |
| Osteoporosis | 106 (10.55) | 13 (17.57) | 0.06 | 13 (17.57) | 13 (17.57) | 1.00 |
| Facture history[¶] | 195 (19.40) | 16 (21.62) | 0.43 | 14 (18.92) | 16 (21.62) | 0.68 |
| Peripheral neuropathy[†] | 42 (4.18) | 1 (1.35) | 0.36 | 6 (8.11) | 1 (1.35) | 0.12 |
| Parkinson's disease[†] | 40 (3.98) | 3 (4.05) | 1.00 | 5 (6.76) | 3 (4.05) | 0.72 |
| Cardiovascular disease | 527 (52.44) | 34 (45.95) | 0.28 | 48 (64.86) | 34 (45.95) | <0.05 |
| Cerebrovascular disease | 156 (15.52) | 10 (13.51) | 0.64 | 14 (18.92) | 10 (13.51) | 0.37 |
| Diabetes mellitus | 639 (63.58) | 44 (59.46) | 0.48 | 49 (66.22) | 44 (59.46) | 0.40 |
| Hypertension | 806 (80.20) | 60 (81.08) | 0.85 | 64 (86.49) | 60 (81.08) | 0.37 |
| Chronic pulmonary disease | 164 (16.32) | 20 (27.03) | <0.05 | 20 (27.03) | 20 (27.03) | 1.00 |
| Cataracts | 249 (24.78) | 18 (24.32) | 0.93 | 20 (27.03) | 18 (24.32) | 0.71 |
| Dementia | 62 (6.17) | 4 (5.41) | 0.79 | 6 (8.11) | 4 (5.41) | 0.51 |
| Mental disorders | 70 (6.97) | 5 (6.76) | 0.95 | 5 (6.76) | 5 (6.76) | 1.00 |
| Rheumatoid arthritis[†] | 17 (1.69) | 2 (2.70) | 0.52 | 1 (1.35) | 2 (2.70) | 1.00 |
| Chronic liver disease | 105 (10.45) | 11 (14.86) | 0.24 | 5 (6.76) | 11 (14.86) | 0.11 |
| Glucocorticoids | 136 (13.53) | 15 (20.27) | 0.11 | 10 (13.51) | 15 (20.27) | 0.27 |
| Antiepileptic drug[†] | 23 (2.29) | 1 (1.35) | 1.00 | 2 (2.70) | 1 (1.35) | 1.00 |
| Anti-depressants | 68 (6.77) | 5 (6.76) | 1.00 | 6 (8.11) | 5 (6.76) | 0.75 |
| Beta-blocking agents | 91 (9.05) | 7 (9.46) | 0.91 | 5 (6.76) | 7 (9.46) | 0.55 |
| Benzodiazepines | 110 (10.95) | 10 (13.51) | 0.50 | 12 (16.22) | 10 (13.51) | 0.64 |
| Analgesics | 101 (10.05) | 11 (14.86) | 0.19 | 5 (6.76) | 11 (14.86) | 0.11 |
| Sedatives and Hypnotics | 151 (15.02) | 10 (13.51) | 0.72 | 14 (18.92) | 10 (13.51) | 0.37 |
| Statin[†] | 82 (8.16) | 5 (6.76) | 0.67 | 3 (4.05) | 5 (6.76) | 0.72 |
| Vitamin D and analogues | 25 (2.49) | 5 (6.76) | <0.05 | 5 (6.76) | 5 (6.76) | 1.00 |
| Diuretics[§] | 91 (9.05) | 9 (12.16) | 0.37 | 7 (9.46) | 9 (12.16) | 0.60 |
| Oral diabetes medications | 140 (13.93) | 12 (16.22) | 0.59 | 10 (13.51) | 12 (16.22) | 0.64 |
| Proton pump inhibitors | 191 (19.00) | 9 (12.16) | 0.14 | 16 (21.62) | 9 (12.16) | 0.12 |
| Nitrates | 48 (4.78) | 4 (5.41) | 0.81 | 6 (8.11) | 4 (5.41) | 0.51 |
| Oral NSAIDs | 158 (15.72) | 11 (14.86) | 0.85 | 18 (24.32) | 11 (14.86) | 0.15 |

Abbreviation: AR, Antiresorptive medications; NSAIDs, Non-steroidal anti-inflammatory drugs.

[¶]Nonvertebral fracture(fractures of radius/ulna, humerus, and other nonvertebral fractures except for hip fracture)

[§]thiazide, loop

[†]statistical test by Fisher's exact test.

fractures before dialysis (N = 1,093), those younger than 50 years (N = 44), those with fracture events in 2004 or 2012 (N = 365), those not found persistently using the drugs (N = 82), and those who had taken other osteoporosis medications (N = 329), we were left with 74 AR users

and 1,005 non-users. After 1:1 propensity score matching, we were left with 148 study participants in the two groups: 74 AR users and 74 non-users (Fig 1).

There was no difference in the distribution of baseline characteristics between AR users and non-users in general. Before matching, the mean age was about 72 years and mean (SD) of duration of dialysis was 2.70 (1.92) years for AR users and 2.90 (2.10) years for AR non-users. AR users were found to have more chronic pulmonary disease, vitamin D and analog use before matching. After matching by propensity score, the distribution of baseline characteristics of the two groups evened out for all the variables except for cardiovascular disease (Table 1).

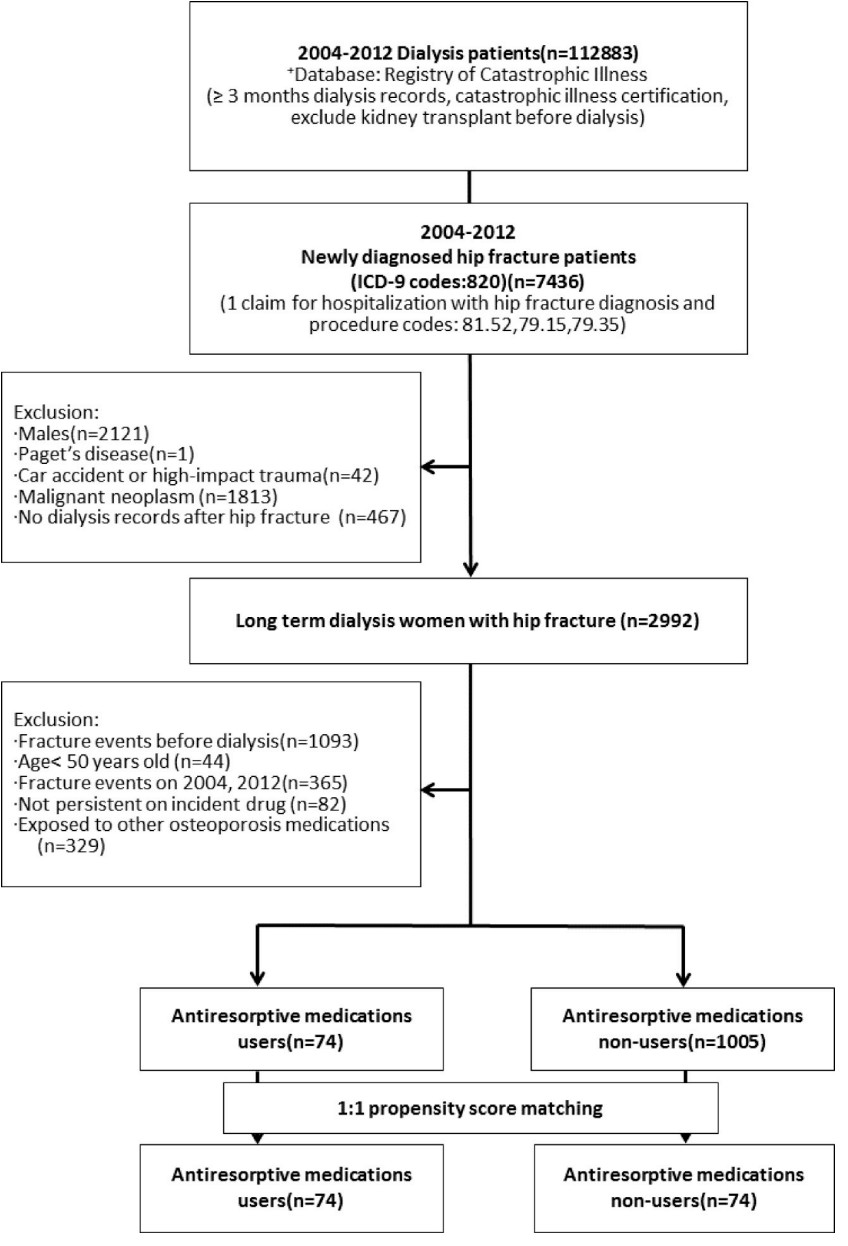

**Fig 1. Study flow diagram.**

### The 5-year cumulative rate and hazard ratio of second hip fracture

No significant difference in 5-year cumulative incidence rates of second hip fracture between AR users and non-users is shown in Fig 2. The cumulative rates of second hip fracture were 17% for AR users and 12% for non-users (Log-rank p-value = 0.95). In multivariable analyses, AR users were found to have insignificant reduction in risk of second hip fracture either using traditionally adjusted [adjusted hazard ratio (aHR) 1.28; 95% confidence interval (CI) 0.57–2.83], PS matching combined with traditionally adjusted (0.60; 0.16–2.20), or traditionally adjusted combined with competing risk of death (0.91; 0.30–2.76) approaches. In addition, the raloxifene users showed insignificant higher risk of developing second hip fracture compared with the alendronate users (ranges of adjusted HR: 2.76–2.80) (Table 2).

### The 2-year cumulative rate and hazard ratio of mortality

Fig 3 exhibits the significantly lower two-year cumulative mortality rate for AR users vis-a-vis non-users (14% vs. 38%, Log-rank p-value = 0.003). Patients who used AR represented significantly lower one-year (aHR 0.28; 95%CI 0.11–0.76) and two-year mortality (0.37, 0.20–0.67) risks compared to those who did not. These results were similar by using PS matching combined with a traditionally adjusted approach. In addition, the multivariable analyses consistently displayed insignificant differences in hazard ratios of one-year and two-year mortality for Raloxifene users compared with Alendronate users (Table 3).

### Sensitivity analyses

The study further explored persistence of AR use and found 84.62% of alendronate users and 73.77% of raloxifene users had received at least six refills, and 46.15% and 19.67% of them had received at least twelve refills (S1 Fig). However, similar results were observed in risks of second hip fracture or mortality between users and non-users and the two groups of AR users regardless of PDC value (S4 Table) or prescription refill records (S5 Table). In addition, the results of risk of second hip fracture and risk of mortality were found consistent in the following analyses: intention-to-treat scenario, excluding patients who switched drugs, extending the

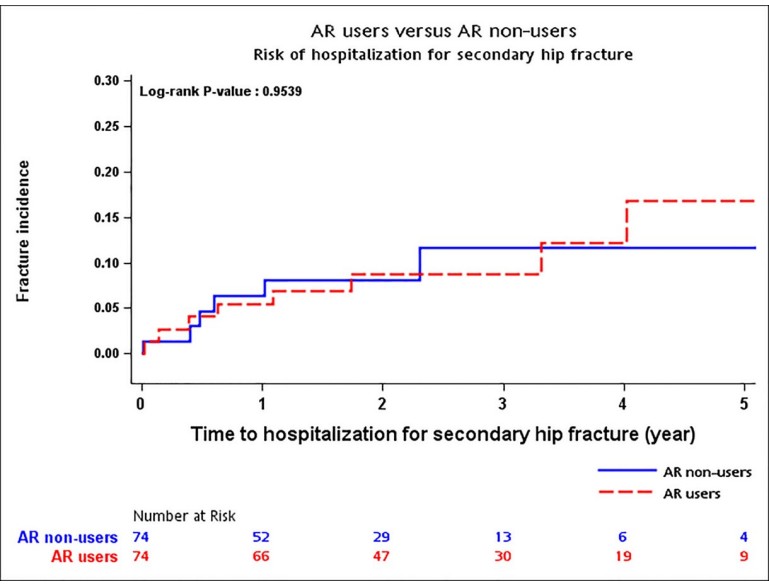

**Fig 2. Cumulative incidence of second hip fracture between AR non-users and AR users.**

**Table 2. Risk of hospitalization for second hip fracture in AR users versus non-users and raloxifene versus alendronate.**

| Outcome | Events, n (%) | Fracture rate, per 100 PYs | Hazard Ratio (95% CI) | | | | | | | |
|---|---|---|---|---|---|---|---|---|---|---|
| | | | Crude | P value | M1 | P value | M2 | P value | M3 | P value |
| AR users versus AR non-users | | | | | | | | | | |
| AR non-users | 56 (5.57) | 2.64 | 1.00 (Reference) | | 1.00 (Reference) | | 1.00 (Reference) | | 1.00 (Reference) | |
| AR users | 8 (10.81) | 3.85 | 1.53 (0.73–3.21) | 0.26 | 1.28 (0.57–2.83) | 0.55 | 0.60 (0.16–2.20) | 0.44 | 0.91 (0.30–2.76) | 0.87 |
| Raloxifene versus Alendronate | | | | | | | | | | |
| Alendronate | 1 (7.69) | 2.91 | 1.00 (Reference) | | 1.00 (Reference) | | 1.00 (Reference) | | 1.00 (Reference) | |
| Raloxifene | 7 (11.48) | 4.22 | 1.38 (0.17–11.32) | 0.76 | 2.76 (0.22–35.30) | 0.44 | 2.76 (0.22–35.30) | 0.43 | 2.80 (0.42–18.79) | 0.29 |

Abbreviation: AR: Antiresorptive medications; PYs: person-years.

M1: Before propensity score matching, adjusted with significant covariates of baseline characteristics in univariate Cox-regression (p<0.1) (S3 Table); M2: Propensity score matching, adjusted with significant covariates of baseline characteristics in univariate Cox-regression (p<0.1) (S3 Table); M3: Adjusted for all variables in M2 and competing risk.

follow-up by 90 days, adjusting for immortal time bias, and in a series of subgroup analyses (S6–S11 Tables).

## Discussion

In this retrospective cohort study, we investigated the effectiveness of AR in preventing hospitalization for second hip fracture and reducing the risk of mortality in women 50 years and older on dialysis with new hip fractures new to antiresorptive therapy. We found no significant differences in the prevention of hospitalization for second hip fracture between those who used AR and those who did not, and no difference in the effectiveness of two AR, raloxifene and alendronate. In addition, the AR users had significantly lower one-year mortality and two-year mortality compared to non-users.

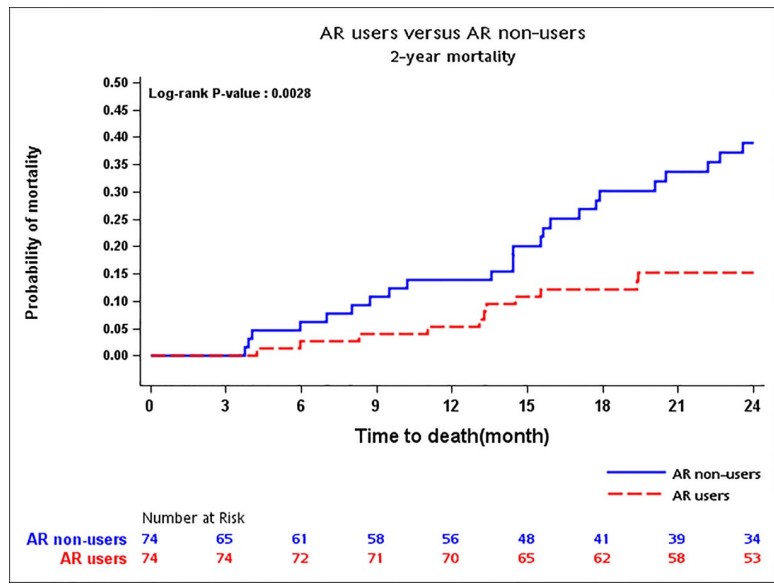

**Fig 3. Two-year cumulative mortality rate between AR non-users and AR users.**

**Table 3. 1-year and 2-year mortalities of AR non-users versus users and raloxifene versus alendronate.**

| Outcome | Events, N (%) | Fracture rate, Per 100 PYs | Hazard Ratio (95% CI)‡ | | | | | |
|---|---|---|---|---|---|---|---|---|
| | | | Crude | P value | M1 | P value | M2 | P value |
| AR users versus AR non-users, 1-year mortality | | | | | | | | |
| AR non-users | 160 (17.84) | 19.10 | 1.00 (Reference) | | 1.00 (Reference) | | 1.00 (Reference) | |
| AR users | 4 (5.41) | 5.52 | 0.28 (0.11–0.76) | <0.05 | 0.26 (0.10–0.70) | <0.05 | 0.25 (0.07–0.90) | <0.05 |
| AR users versus AR non-users, 2-year mortality | | | | | | | | |
| AR non-users | 313 (34.89) | 21.96 | 1.00 (Reference) | | 1.00 (Reference) | | 1.00 (Reference) | |
| AR users | 11 (14.86) | 8.24 | 0.37 (0.20–0.67) | <0.05 | 0.36 (0.20–0.65) | <0.05 | 0.35 (0.17–0.72) | <0.05 |
| Raloxifene versus Alendronate, 1-year mortality | | | | | | | | |
| Alendronate | 1 (7.69) | 8.31 | 1.00 (Reference) | | 1.00 (Reference) | | 1.00 (Reference) | |
| Raloxifene | 3 (4.92) | 5.05 | 0.61 (0.06–5.85) | 0.67 | 0.80 (0.08–8.17) | 0.85 | 0.80 (0.08–8.17) | 0.85 |
| Raloxifene versus Alendronate, 2-year mortality | | | | | | | | |
| Alendronate | 1 (7.69) | 4.34 | 1.00 (Reference) | | 1.00 (Reference) | | 1.00 (Reference) | |
| Raloxifene | 10 (16.39) | 9.43 | 2.12 (0.27–16.60) | 0.47 | 1.83 (0.23–14.41) | 0.57 | 1.83 (0.23–14.41) | 0.57 |

Abbreviation: AR: Antiresorptive medications; PYs: person-years. Notes: M1: Before propensity score matching, adjusted with significant covariates of baseline characteristics in univariate Cox-regression (p<0.1) (S3 Table); M2: Propensity score matching, adjusted with significant covariates of baseline characteristics in univariate Cox-regression (p<0.1) (S3 Table)

‡Time-varying adjusted failure.

Although AR users tended to have slightly lower risk of hospitalization for second hip fracture than non-users, our results reflect that there were no obvious effects of AR in preventing second hip fracture. There are several explanations for these observations. Firstly, a low number of events were observed, thus leading to inaccurate estimated effect size. Secondly, reduced renal function itself might lead to lower bone mineral density, then offset effects of AR. Thirdly, the impact of chronic kidney disease (CKD)-mineral and bone disorder that appears quite common in dialysis patients [22, 23] might also be a factor that confounds the effects of the AR, especially in dialysis women with higher probability for uremic bone disease and postmenopausal osteoporosis compared to women not on dialysis [24]. Several studies have demonstrated that AR might exacerbate the risk of adynamic bone disease (low turnover bone disease) [25–27]; however, determining the stage of bone disease in need of an invasive examination of bone biopsy [28] is challenging to implement in clinical practice. Not only do patients have lower willingness to receive an invasive examination but bone biopsy requires experts to interpret the results as well. Furthermore, bone biopsy can only reflect the bone mass at a single time, and it cannot represent the whole body of the bone, and additionally, the measurement does not take into account the CKD stage and the time of disease [28, 29]. As a consequence, clear consensus on using AR drugs in the dialysis population for second hip fracture prevention is still lacking, but in spite of this lack of obvious benefit on fracture prevention, our results shed light on exploring potential mechanisms (conditions) explaining the association between AR and refracture in the dialysis population.

The results from previous studies supported our observation that AR use is associated with significant risk reduction of mortality [30–33]. The mechanisms of AR on mortality risk reduction is complex and also involves drug actions and their interactions with patient conditions. Cardiovascular disease as highly caused by vascular calcification and metabolic problems has been considered the primary cause of death in patients with renal disease [34]. Previous studies have demonstrated that bisphosphonates might inhibit or reduce vascular calcification in dialysis patients [35–37], although a recent post-hoc analysis of the HORIZON Pivotal Fracture Trial failed to support the hypothesis that zoledronic acid could affect the formation or

progression of vascular calcification in postmenopausal women with osteoporosis [38]. This discrepancy might reflect drug actions of different classes of bisphosphonates, where nitrogen-containing bisphosphonates for example have much stronger AR potency without inhibiting bone mineralization than non-nitrogen-containing ones, and might have indirect effects on vascular calcification through influencing serum levels of calcium, phosphate, and parathyroid hormone. These actions imply possible benefits of nitrogen-containing bisphosphonates (e.g., alendronate) on mortality reduction in dialysis patients. In addition, although the post-hoc analyses from Multiple Outcomes of Raloxifene Evaluation manifested a neutral effect of raloxifene on the cardiovascular events in general patients [39], raloxifene has been shown to possess promise for reduction of serum lipids in patients such as our study population [6, 40, 41]. More research with larger sample sizes in exploring effects of types of AR on vascular calcification, biochemical actions, cardiovascular disease, and mortality in patients on hemodialysis is obviously needed.

Higher persistence rates of AR use might also be associated with lower refracture and mortality risks. Current results similar to Siris and Grangers's findings showed patients who refilled more times or had more PDCs of AR tended to be at less risk of second hip fracture and mortality than non-users [42, 43]. Persistence with AR therapy might be influenced by multidimensional causes; for example, the health system, personality characteristics such as health beliefs, comorbidity, tolerability, medication dosing and adverse events, or any of these in combination with each other [44–46]. Despite the underlying mechanisms being complex and beyond the scope of the study, our observations encourage further study to elucidate the effects of adherence and persistence with AR therapy on fracture prevention in this fragile population.

Although several extensive analyses to reduce bias have been performed in the study, some limitations should be declared. Firstly, the sample size was small, though it was representative, and the information in Taiwan's dialysis database is relatively more complete than that found in other countries. Secondly, data representing potential confounding factors such as smoking, caffeine intake, lifestyle, family history, and laboratory data were not available in the claim data. Thirdly, lack of detailed and accurate information on etiology of CKD and use of glucocorticoid made it problematic to explore any influences of these on the results. Fourthly, novel AR drugs like RANKL inhibitors, which were launched in Taiwan in 2011, were not included in the study through limitation of data resources (from 2004 to 2012). Fifthly, because the database used in this study could not provide cause of mortality in detail, the effect of AR on risk of specific mortality could not be further evaluated. Finally, the study concerns the dialysis population of Taiwan, so generalizing results to other races, ethnic groups, or countries should be cautioned.

## Conclusions

In conclusion, AR did not significantly improve the risk of second hip fracture compared to those not using AR; however, AR users were found to have lower mortality rates than non-users. Further larger studies are needed to evaluate effectiveness of AR on clinical outcomes in dialysis.

## Supporting information

**S1 Fig. Persistence rates of alendronate and raloxifene.**
(DOCX)

**S1 Table. Corresponding ICD-9-CM codes used for the chronic conditions/comorbidities in this study.**
(DOCX)

**S2 Table. Corresponding ATC codes used for the comedications in this study.**
(DOCX)

**S3 Table. Significant covariates of baseline characteristics in univariate Cox-regression.**
(DOCX)

**S4 Table. The PDC stratification and outcomes between AR users/non-users and treatment group.**
(DOCX)

**S5 Table. The refilled times stratification and outcomes between AR users/non-users and treatment group.**
(DOCX)

**S6 Table. Subgroup analysis of outcomes between AR users/non-users and alendronate/raloxifene.**
(DOCX)

**S7 Table. Sensitivity analysis of hospitalization for second hip fracture: Excluding short-term users.**
(DOCX)

**S8 Table. Sensitivity analysis: Intention-to-treat scenario.**
(DOCX)

**S9 Table. Sensitivity analysis: Non-change group.**
(DOCX)

**S10 Table. Sensitivity analysis: Primary analysis plus 90 days.**
(DOCX)

**S11 Table. Sensitivity analysis: Immortal time and lag time adjustment.**
(DOCX)

## Acknowledgments

We would like to thank Professor Steve Tredrea for editing this manuscript.

## Author Contributions

**Conceptualization:** Tien-Ching Lee, Shun-Jin Lin.

**Data curation:** Shun-Jin Lin.

**Formal analysis:** Yu-Ciou Lin, Chung-Yu Chen.

**Funding acquisition:** Shang-Jyh Hwang, Ming-Yen Lin.

**Methodology:** Yu-Ciou Lin, Tien-Ching Lee, Chung-Yu Chen, Shang-Jyh Hwang, Ming-Yen Lin.

**Resources:** Shang-Jyh Hwang, Ming-Yen Lin.

**Supervision:** Shang-Jyh Hwang.

**Validation:** Tien-Ching Lee.

**Writing – original draft:** Yu-Ciou Lin.

**Writing – review & editing:** Tien-Ching Lee, Chung-Yu Chen, Shun-Jin Lin, Shang-Jyh Hwang, Ming-Yen Lin.

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
