## [Decision Letter · Decision Letter 0]

5 Jun 2020

PONE-D-20-07988

Effectiveness of antiresorptive medications in women on long-term dialysis after hip fracture: A population-based cohort study

PLOS ONE

Dear Dr. Lin,

Thank you for submitting your manuscript to PLOS ONE. After careful consideration, we feel that it has merit but does not fully meet PLOS ONE’s publication criteria as it currently stands. Therefore, we invite you to submit a revised version of the manuscript that addresses the points raised during the review process.

We look forward to receiving your revised manuscript.

Kind regards,

Yuanyuan Wang

Academic Editor

PLOS ONE

Journal Requirements:

Additional Editor Comments (if provided):

This study answered an important clinical question and provided some insight into the potential benefits of antiresorptive medications on outcomes in dialysis patients. The reviewers have raised concerns about the methodology of the study and requested further justification of the statistical analysis. The manuscript requires English language revision.

Reviewers' comments:

Reviewer's Responses to Questions

**Comments to the Author**

1. Is the manuscript technically sound, and do the data support the conclusions?

Reviewer #1: Yes

Reviewer #2: Yes

Reviewer #3: Partly

2. Has the statistical analysis been performed appropriately and rigorously? 

Reviewer #1: Yes

Reviewer #2: Yes

Reviewer #3: Yes

3. Have the authors made all data underlying the findings in their manuscript fully available?

Reviewer #1: Yes

Reviewer #2: No

Reviewer #3: Yes

4. Is the manuscript presented in an intelligible fashion and written in standard English?

Reviewer #1: Yes

Reviewer #2: Yes

Reviewer #3: No

5. Review Comments to the Author

Reviewer #1: This paper represents a good attempt at answering real world clinical questions about what benefits antiresorptive medications have on outcomes in dialysis patients, who are often excluded from trials of these agents.

The biggest issue I have is the small numbers despite the very large populations. In particular, why was the matching done on a 1:1 basis? I understand why the AR patients are limited, but the matched cases could be 1:2 or even 1:3. Effect sizes are large enough that differences in mortality are statistically significant regardless (Table 3), but the effects in Table 2 look underpowered, although I agree that if the M3 analyses are considered the "best" model, this will have no effect on the final result.

I think this paper would be improved by increasing the number of matched controls, but I understand this would add a lot of work for the authors. I am particularly interested in an explanation of how the authors came to a decision as to how many matches were appropriate, particularly if they can justify the 1:1 matching or not.

Regarding mechanisms, a recent paper suggests that any mechanisms of bisphosphonate use on mortality are likely to be independent of vascular calcification, at least for zoledronic acid, and as such this is worth adding to the discussion (paragraph 3): Cai G, Keen HI Host L, Laslett LL, Aitken D, Winzenberg T, Wluka AE, Black DM, Jones G. Once-yearly zoledronic acid and change in abdominal aortic calcification over 3 years in postmenopausal women with osteoporosis: results from the HORIZON Pivotal Fractural Trial[in press]. Osteoporos Int 2020 Accepted 22 April 2020 doi: 10.1001/jama.2020.2938

Minor issue: edit the word "morality" in the abstract - should be mortality.

Reviewer #2: The authors conduct a retrospective cohort study to compare the risk of hospitalization for secondary hip fracture and risk of mortality between antiresorptive (AR) drugs and non-user group for long-term dialysis women. 74 AR users and 74 non-users were analyzed from Taiwan National Health Insurance Research Datasets. The results showed that alendronate user have higher risk of secondary hip fracture compared to the raloxifene users. AR users showed lower mortalities than non-users. They concluded that AR treatment didn’t improve the risk of secondary hip fracture but had reduced mortality for dialysis women.

1. Propensity score matching was used in this work. However, it lacks any detail information on what factors were considered and how it was implemented.

2. Some languages need to be clarified. For example, the following sentences sounds conflict. “Because the group may change over time, …”, Patients were censored if they switched to other treatment groups once treatment began”, “Patients who switched drugs during the study period were excluded”

3. The authors excluded male patients (N=2121) in this study. The sample size seems comparable between two sexes. Is there any reason to justify why male sample wasn’t considered?

4. The prevalence of chronic heart disease is significantly different at baseline between users and non-users. How will this affect the final conclusion especially for the mortality analysis. was the cause of mortality taken into consideration?

5. Please clearly spell out the covariates adjusted in the model, either in the statistical analysis section or the relevant results section, rather than just saying adjusting for the significant covariates in univariate analysis…

Reviewer #3: This manuscript examines the effect of ant-fracture therapy on second and subsequent hip fractures and mortality in a cohort of dialysis patients. The data presented are reminiscent to the landmark Lyles study that demonstrated a mortality benefit of zoledronic acid but failed to demonstrate a significant reduction in second or subsequent hip fractures when administered after hip fracture. This is an extremely valuable dataset given the dearth of information about osteoporosis treatments in patients with stage 5 CKD but is clearly underpowered for hip fracture outcomes after propensity matching with only 74 individuals in the treatment and non-treatment arms. In this regard, the exclusion of participants using anti-resorptive therapies other than alendronate and raloxifene is puzzling and would appear only to serve to reduce the power of the study. Furthermore, the decision to exclude participants that change therapies exacerbates this problem. On the other hand the decision to include raloxifene, a weak anti-resorptive without proven anti-hip fracture efficacy in the analysis is problematic. In order to maximise power, these participants should probably remain with a sensitivity analysis excluding them as part of the exploratory analysis. A comparison of these outcomes, as already presented in the manuscript, between the more potent anti-resportive agents and raloxifene would remain of interest. Given that this is a postmenopausal population, was there any menopausal hormone therapy use and could such individuals be added to the antiresorptive treated participants with a view to increasing power? The exclusion of those treated to bone formation stimulating agents or changed to bone formation stimulating agents is appropriate. Exposure to glucocorticoids, past and current, adds another layer of complexity to this analysis, given the difference in the pathogenesis of osteoporosis in glucocorticoid-treated patients and postmenopausal osteoporosis as well as the likelihood that up to a third of participants may have had prior glucocorticoid exposure for treatment of glomerulonephritis. Does the data permit any examination of past glucocorticoid exposure and contain a breakdown of the aetiology of CKD?

The discussion would be enhanced by discussion of the complexities involved in the differentiation of osteoporosis from CKDMB and acknowledging concerns that use of potent anti-resorptive might increase the risk of adynamic bone disease. Further, reference should be made to post-hoc analyses of the MORE study that demonstrated a neutral effect of raloxifene on cardiovascular outcome with the possible exception of fatal stroke.

Other points:

1. the description of the ascertainment of mortality is unclear and leads me to question whether ascertainment was complete.

2. the definition of adherence and persistence to therapy should be provided in methods.

3. participants were stratified by income, this is not the same as socioeconomic status

4. how was osteoporosis defined?

5. how was chronic heart disease ascertained/defined

6. diuretic use was presumably loop diuretic use though any thiazide use that could impact on calcium homeostasis should be identified

7. secondary hip fracture should be changed to second or subsequent hip fracture throughout the manuscript.

6. PLOS authors have the option to publish the peer review history of their article (what does this mean?). If published, this will include your full peer review and any attached files.

Reviewer #1: No

Reviewer #2: No

Reviewer #3: No

---

## [Author Response · Author response to Decision Letter 0]

16 Jul 2020

Reviewer #1: This paper represents a good attempt at answering real world clinical questions about what benefits antiresorptive medications have on outcomes in dialysis patients, who are often excluded from trials of these agents.

1. The biggest issue I have is the small numbers despite the very large populations. In particular, why was the matching done on a 1:1 basis? I understand why the AR patients are limited, but the matched cases could be 1:2 or even 1:3. Effect sizes are large enough that differences in mortality are statistically significant regardless (Table 3), but the effects in Table 2 look underpowered, although I agree that if the M3 analyses are considered the "best" model, this will have no effect on the final result. I think this paper would be improved by increasing the number of matched controls, but I understand this would add a lot of work for the authors. I am particularly interested in an explanation of how the authors came to a decision as to how many matches were appropriate, particularly if they can justify the 1:1 matching or not.

Ans: Thank you for valuable comment. Eventually, we had tried to increase pair of matched control (1:2-1:5) to enhanced our statistical power. However, a serious case number lost in AR group happened when increasing pair of matched control. We consider that cases of AR group may provide more useful information than those of Non-AR group and should be preserved as possible as we could. Therefore, we keep 1:1 matching approach to maintain valuable information from AR group. We believe that statistical power may not be the main influence on estimation of effect size. In addition, we have added one sentence to clarify this point. 

On line 2-7, page 9: 

“Then, propensity score of AR use was calculated by considering all covariates listed in table 1 as independent variables through multiple logistic regression with forced entry approach. To possibly obtain sufficient statistical power, PS matching pair from 1:1 to 1:5 using the Greedy 5 to 1 digit technique [15] was taken into consideration. Finally, 1:1 matching pair with similar propensity scores was chosen to maintain largest case number of AR use group.” 

2. Regarding mechanisms, a recent paper suggests that any mechanisms of bisphosphonate use on mortality are likely to be independent of vascular calcification, at least for zoledronic acid, and as such this is worth adding to the discussion (paragraph 3): Cai G, Keen HI Host L, Laslett LL, Aitken D, Winzenberg T, Wluka AE, Black DM, Jones G. Once-yearly zoledronic acid and change in abdominal aortic calcification over 3 years in postmenopausal women with osteoporosis: results from the HORIZON Pivotal Fractural Trial[in press]. Osteoporos Int 2020 Accepted 22 April 2020 doi: 10.1001/jama.2020.2938

Ans: Thank you for valuable comment. We have added this in our discussion in new edition to describe possible mechanisms of bisphosphonate use on mortality.

On line 10-21, page 23:

“Previous studies have been demonstrated that bisphosphonates might inhibit or reduce vascular calcification in dialysis patients [35-37]. However, a recent post-hoc analysis of the HORIZON Pivotal Fracture Trial failed to support that zoledronic acid could affect the formation or progression of vascular calcification in postmenopausal women with osteoporosis [38]. This discrepancy may reflect drug actions of different classes of bisphosphonates. Nitrogen-containing bisphosphonates have much stronger AR potency without inhibiting bone mineralization than non-nitrogen-containing ones, and might have indirect effects on vascular calcification through influencing serum levels of calcium, phosphate, and parathyroid hormone. These actions imply possible benefits of nitrogen-containing bisphosphonates (e.g., alendronate) on mortality reduction in dialysis patients.”

3. Minor issue: edit the word "morality" in the abstract - should be mortality.

Ans: Sorry for this typing error and thank you for pointing out the error. We have corrected this error in the new version. 

Reviewer #2: The authors conduct a retrospective cohort study to compare the risk of hospitalization for secondary hip fracture and risk of mortality between antiresorptive (AR) drugs and non-user group for long-term dialysis women. 74 AR users and 74 non-users were analyzed from Taiwan National Health Insurance Research Datasets. The results showed that alendronate user have higher risk of secondary hip fracture compared to the raloxifene users. AR users showed lower mortalities than non-users. They concluded that AR treatment didn’t improve the risk of secondary hip fracture but had reduced mortality for dialysis women.

1. Propensity score matching was used in this work. However, it lacks any detail information on what factors were considered and how it was implemented. 

Ans: Sorry for this incomplete description. We have modified this sentence to be clear and readable. 

On line 2-6, page 9: 

“Then, propensity score of AR use was calculated by considering all covariates listed in table 1 as independent variables through multiple logistic regression with forced entry approach. To possibly obtain sufficient statistical power, PS matching pair from 1:1 to 1:5 was taken into consideration. Finally, 1:1 matching pair with similar propensity scores using the Greedy 5- to 1- digit technique [15] was chosen to maintain largest case number of AR use group.” 

2. Some languages need to be clarified. For example, the following sentences sounds conflict. “Because the group may change over time, …”, Patients were censored if they switched to other treatment groups once treatment began”, “Patients who switched drugs during the study period were excluded” 

Ans: Sorry for the unclear description. We modified the relevant descriptions in the updated version. Our primary analysis used the on-treatment scenario, that is, patients were censored if they switched to other osteoporosis treatment once AR treatment began. The covariates were adopted as time-dependent factors for adjusting different follow-up periods. We further excluded these patients in sensitivity analysis to confirm the exact drug effect. 

On line10-12, page 9: 

“Because patients who switched to other osteoporosis medications were considered censored, the covariates were adopted as time-dependent factors for adjusting different follow-up periods.”

On line 23-24, page 9 and line 1, page 10:

 “The patients who switched to other osteoporosis medications were considered censored in the primary analysis. We further excluded these patients in sensitivity analysis to confirm the exact drug effect.”

3. The authors excluded male patients (N=2121) in this study. The sample size seems comparable between two sexes. Is there any reason to justify why male sample wasn’t considered?

Ans: We only included female patients because below reasons. First, female is more vulnerable to fractures, which might allow us obtain more cases of second fractures. Previous study has been demonstrated that a second hip fracture incidence rate was 206 per 10,000 person-years for women and 174 per 10,000 person-years for men.1 Although women have higher refracture risk than men, the mortality rate is twice in men compared with women within the first year after fracture.2,3 Then, female rather than male is more appropriate to be used in exploring effectiveness of AR. The pathology of osteoporosis is different between women and men. Postmenopausal osteoporosis affects primarily trabecular bone in the decade following menopause, with fractures occurring predominantly at vertebral and distal forearm sites. Estrogen deficiency increases bone resorption more than formation. However, men’s bones have a mechanical advantage because the larger bone diameter makes them more fracture-resistant.4 Furthermore, the indication of raloxifene is to prevent and treat postmenopausal osteoporosis. To synthesize above considerations and avoid unmeasured confounding bias that may interfere with data integrity and evaluate the exact treatment effect on dialysis patients with hip fractures. We considered postmenopausal women who undergone dialysis as our study population.

Reference 

1.Incidence of second hip fractures and associated mortality in Taiwan: A nationwide population-based study of 95,484 patients during 2006e2010 

2.Excess Mortality in Men Compared With Women Following a Hip Fracture. National Analysis of Comedications, Comorbidity and Survival. PMID: 20075035 DOI: 10.1093/ageing/afp221

3.Gender Differences in Mortality After Hip Fracture: The Role of Infection. PMID: 14672359 DOI: 10.1359/jbmr.2003.18.12.2231

4.Joseph T. DiPiro RLT, Gary C. Yee, et al. Pharmacotherapy: A Pathophysiologic Approach, 9e. 

4. The prevalence of chronic heart disease is significantly different at baseline between users and non-users. How will this affect the final conclusion especially for the mortality analysis. Was the cause of mortality taken into consideration?

Ans: For consistency, we changed the term “chronic heart disease” with “cardiovascular disease”. We agree that the presence of cardiovascular disease may possibly confound the analysis of mortality. Thus, we conducted a series analyses to investigate or reduce the influence of cardiovascular disease. In general, similar results were found after controlling the influence of cardiovascular disease on mortality (Table 3). In addition, we stratified by cardiovascular disease in Table S6, which shows the trend of lower mortality rate in AR user, is consistent for our conclusion. The anti-resorptive medication use in AR group did not significantly decrease the incidence of second hip fracture but reduce mortality substantially in older women on dialysis. 

The NHIRD we used in this study cannot provide the cause of mortality in detail. We also add this description in the section of the limitation of this study. 

On line 20-22, page 23: 

“Fifth, because the database we used in this study cannot provide the cause of mortality in detail, the effect of AR on risk of specific mortality could not be further evaluated.”

5. Please clearly spell out the covariates adjusted in the model, either in the statistical analysis section or the relevant results section, rather than just saying adjusting for the significant covariates in univariate analysis…

Ans: Thank you for the valuable suggestion. Because significant covariates of baseline characteristics in univariate Cox-regression are varied in different outcome measures, we have added significant covariates of baseline characteristics in univariate Cox-regression (P<0.1) for risks of second fracture, and different observed period of mortality in Supporting information (S3 Table) and responding sentences in the manuscript. 

On line 1-2, page 9: 

“First of all, we used multivariable regression models to adjusted for significant covariates with criterion P value <0.1 in univariable Cox-regression (S3 Table).” 

Reviewer #3: This manuscript examines the effect of ant-fracture therapy on second and subsequent hip fractures and mortality in a cohort of dialysis patients. The data presented are reminiscent to the landmark Lyles study that demonstrated a mortality benefit of zoledronic acid but failed to demonstrate a significant reduction in second or subsequent hip fractures when administered after hip fracture. This is an extremely valuable dataset given the dearth of information about osteoporosis treatments in patients with stage 5 CKD but is clearly underpowered for hip fracture outcomes after propensity matching with only 74 individuals in the treatment and non-treatment arms. 

1. In this regard, the exclusion of participants using anti-resorptive therapies other than alendronate and raloxifene is puzzling and would appear only to serve to reduce the power of the study. Furthermore, the decision to exclude participants that change therapies exacerbates this problem. On the other hand, the decision to include raloxifene, a weak anti-resorptive without proven anti-hip fracture efficacy in the analysis is problematic. In order to maximise power, these participants should probably remain with a sensitivity analysis excluding them as part of the exploratory analysis. 

Ans: Thank you for your precious suggestions. We agree that study subjects should be preserved as possible as we can. Eventually, decision for patient exclusion in this study is a compromise between ideal condition and real-world situations. In Taiwan, the NHI programs cover anti-resorptive therapies, including bisphosphonates, SERM (e.g., raloxifene), and RANKL inhibitor (denosumab). The denosumab is covered by the NHI program in 2012, which is over the study period (the study period is from 2004 to 2011 limited by the data resource). We also added on the relevant description in the limitation section: “Fourth, the novel anti-resorptive drugs like RANKL inhibitors, which launched in Taiwan since 2011, is not included in our study limited by data resources (from 2004 to 2012)”. Other common anti-resorptive drug is calcitonin. In 2013, TFDA abolished the indication of calcitonin on the postmenopausal osteoporosis treatment because of higher safety risks and uncertain benefits on the population. Thus we did not take calcitonin as the treatment group (see Table R1). In practical, quite low percentages (<15 patients during the whole study period) in our population used other bisphosphates, including zoledronic acid, ibandronate, clodronate, pamidronate, and risedronate. Our primary analysis used the on-treatment scenario; that is, patients were censored if they switched to other osteoporosis treatment once AR treatment began. The on-treatment scenario considers the treatments patients received, and the possibility of the drug changed in reality and thus reduce the bias from the drug exchanged.1 In the sensitivity analysis, we change the scenario to intention-to-treat. Specifically, drug group was classified as the earliest exposed drug and analyzed based on the assumption that patients’ exposure to medications continued to end of follow-up. The results were consistent with the primary analysis. Besides, the drug-change patients account for quite low in the study population. One subject (7.49%) in the alendronate group, and four subjects (6.56%) in the raloxifene group. Regarding the influence of drug changes, we excluded these patients in sensitivity analysis to confirm the exact drug effect.

Reference 

1. Chene G, Morlat P, Leport C, et al. Intention-to-treat vs. on-treatment analyses of clinical trial data: experience from a study of pyrimethamine in the primary prophylaxis of toxoplasmosis in HIV-infected patients. ANRS 005/ACTG 154 Trial Group. Controlled clinical trials. 1998;19(3):233-48.

2. A comparison of these outcomes, as already presented in the manuscript, between the more potent anti-resportive agents and raloxifene would remain of interest. Given that this is a postmenopausal population, was there any menopausal hormone therapy use and could such individuals be added to the antiresorptive treated participants with a view to increasing power? 

Ans: Thanks for the valuable advice. Indeed, hormone replacement therapy (HRT) in the study population had been considered in our analysis. As blow table shown, the percentages of HRT are quite low for AR users groups. Therefore, we did not include the covariate in the analysis.

3. The exclusion of those treated to bone formation stimulating agents or changed to bone formation stimulating agents is appropriate. Exposure to glucocorticoids, past and current, adds another layer of complexity to this analysis, given the difference in the pathogenesis of osteoporosis in glucocorticoid-treated patients and postmenopausal osteoporosis as well as the likelihood that up to a third of participants may have had prior glucocorticoid exposure for treatment of glomerulonephritis. Does the data permit any examination of past glucocorticoid exposure and contain a breakdown of the aetiology of CKD?

Ans: Thank you for the valuable suggestions. We agree that exposure to glucocorticoids and aetiology of CKD were both important factors influencing the effectiveness of AR medications. We have not further carried out these relevant analyses due to below reasons. First, although diabetes mellitus should be the major cause of CKD in Taiwan, information of renal pathology is usually lacking because only small proportion of patients with CKD was willing to receive renal biopsy. Second, accurate information of cause of CKD cannot be available from claim data. Third, patients with CKD could obtain over-the-counter glucocorticoid from the health insurance coverage. Therefore, we only included them as comorbidity and comedicine to recent analyses. In addition, we added this valuable point into our study limitations, “Third, lack of detailed and accurate information on etiology of CKD and use of glucocorticoid make us be not able to completed explore influences of them on our results.” to reflect its importance.

4. The discussion would be enhanced by discussion of the complexities involved in the differentiation of osteoporosis from CKDMB and acknowledging concerns that use of potent anti-resorptive might increase the risk of a dynamic bone disease. Further, reference should be made to post-hoc analyses of the MORE study that demonstrated a neutral effect of raloxifene on cardiovascular outcome with the possible exception of fatal stroke.

Ans: Thanks for the precious suggestions. We have added some sentences in discussion to demonstrate the complexities in the differentiation of osteoporosis from CKDMB and acknowledging concern the risk of a dynamic bone disease in anti-resorptive drug use. Besides, we also added the results of post-hoc analyses of MORE study in the discussion paragraph to investigate the relationship between raloxifene and cardiovascular events.

On line 17-24, page 21 and line 1-4, page 22: 

“Several studies demonstrated that AR might exacerbate the risk of adynamic bone disease (low turnover bone disease) [25-27]. However, to determine the stage of bone disease in need of an invasive examination of bone biopsy [28], it is challenging to implement in clinical practice. Not only patients have a lower willingness to receive an invasive examination, but bone biopsy requires experts to interpret the results. Furthermore, bone biopsy can only reflect the bone mass at a single time, and it cannot represent the whole body of the bone. In addition, the measurement does not take into account the CKD stage and the time of disease [28, 29]. Therefore, there still lacks a clear consensus on using of AR drugs in the dialysis population for second hip fracture prevention. In spite of the lack of obvious benefit on fracture prevention, our results shed a light on exploring potential mechanisms (conditions) explained the association between AR and refracture in the dialysis population.”

On line 19-23, page 22 and line 1-2, page 23: 

“In addition, although the post-hoc analyses from Multiple Outcomes of Raloxifene Evaluation manifested a neutral effect of raloxifene on the cardiovascular events in general patients [39], raloxifene has been shown some promise for reduction of serum lipids in patients like our study population [6, 40, 41]. However, more researches with larger sample size in exploring effects of types of AR on vascular calcification, biochemical actions, cardiovascular disease, and mortality in patients on hemodialysis is needed.”

Other points:

5-1 the description of the ascertainment of mortality is unclear and leads me to question whether ascertainment was complete.

Ans: We apologized for the inaccurate description. We have revised the sentence as “Death was defined as death during hospitalization or patients discharged in critical conditions and disenrolled in NHI within 3 days after discharge.” 

5-2 the definition of adherence and persistence to therapy should be provided in methods.

Ans: Thanks for the valuable suggestions. We have added the definition of adherence and persistence in the method section.

On line 15-19, page 9:

“The PDC was calculated by the sum of unique days with AR supply divided total number of days in the observed period. The persistence was defined by AR prescription refilled within 30 days between the last prescription date of AR and the next prescription [19, 20]. We stratified AR users by their times of prescription refills (≥3, ≥6, and ≥9 times) to estimate changes of main results.”

5-3 participants were stratified by income, this is not the same as socioeconomic status

Ans: We have correct this word to be consistent word, socioeconomic status. Some literature points out that insurance amount represents the research object's social and economic status.1,2 In this study, the insurance file of the ID file (INS_AMT) is retrieved as the representative of the socioeconomic situation. Definition: (1) Low socioeconomic status means that the insurance amounts are less than NT$15 840 (2) The middle socioeconomic status is insurance amounts ranging from NT$15841 to NT$25000 (3) High socioeconomic status means that the insurance amounts are greater than NT$25001. Since Taiwan’s 2006 government stipulated that the minimum salary for full-time employees is NT$ 15840, NT$ 15840 is the cut-off point for minimum income in the study.

Reference

1. Lin HC, Chao PZ, Lee HC. Sudden sensorineural hearing loss increases the risk of stroke: a 5-year follow-up study. Stroke. 2008;39(10):2744-8.

2. Chen CH, Huang KY, Wang JY, et al. Combined effect of individual and neighbourhood socioeconomic status on mortality of rheumatoid arthritis patients under universal health care coverage system. Fam Pract. 2015;32(1):41-8.

5-4 how was osteoporosis defined?

Ans: Osteoporosis was defined based on ICD-9-CM codes (733.0) listed on ≥ 2 ambulatory care claims records or ≥ 1 inpatient care claims records during the year prior to the index date. We have also added several sentences on method and 2 tables on supplement. 

On line 4-9, page 8: 

“Osteoporosis, fracture history, and comorbidity were defined based on ICD-9-CM codes (S1 Table) listed on ≥ 2 ambulatory care claims records or ≥ 1 inpatient care claims records during the year prior to the index date. The listed co-medicines were identified based on Anatomical Therapeutic and Chemical codes (S2 Table). Patients were co-medication users when listed medicines prescribed over 28 defined daily dose (DDD).”

5-5 how was chronic heart disease ascertained/defined?

Ans: For consistency, we changed the term “chronic heart disease” as “cardiovascular disease”. We have added detailed definitions of comorbidity and comedicine in new edition and new S1-S2 tables. The cardiovascular disease includes myocardial infarction, chronic heart failure, and arrhythmia. (ATC code: 410-414, 410.XX, 428, 426-427)

On line 4-9, page 8: 

“Osteoporosis, fracture history, and comorbidity were defined based on ICD-9-CM codes (S1 Table) listed on ≥ 2 ambulatory care claims records or ≥ 1 inpatient care claims records during the year prior to the index date. The listed co-medicines were identified based on Anatomical Therapeutic and Chemical codes (S2 Table). Patients were co-medication users when listed medicines prescribed over 28 defined daily dose (DDD).”

5-6 diuretic use was presumably loop diuretic use though any thiazide use that could impact on calcium homeostasis should be identified

Ans: Thanks for the valuable suggestions. In the study, the covariate of diuretic use is including loop and thiazide (S2 table. Corresponding ATC codes used for the comedications in this study).

5-7 secondary hip fracture should be changed to second or subsequent hip fracture throughout the manuscript.

Ans: Thank you for correcting. We have changed it to second hip fracture throughout the manuscript.

---

## [Decision Letter · Decision Letter 1]

6 Aug 2020

PONE-D-20-07988R1

Effectiveness of antiresorptive medications in women on long-term dialysis after hip fracture: A population-based cohort study

PLOS ONE

Dear Dr. Lin,

Thank you for submitting your manuscript to PLOS ONE. After careful consideration, we feel that it has merit but does not fully meet PLOS ONE’s publication criteria as it currently stands. Therefore, we invite you to submit a revised version of the manuscript that addresses the points raised during the review process.

We look forward to receiving your revised manuscript.

Kind regards,

Yuanyuan Wang

Academic Editor

PLOS ONE

Additional Editor Comments (if provided):

The reviewers' comments have been addressed appropriately. However, the manuscript will require considerable language editing.

Reviewers' comments:

Reviewer's Responses to Questions

**Comments to the Author**

1. If the authors have adequately addressed your comments raised in a previous round of review and you feel that this manuscript is now acceptable for publication, you may indicate that here to bypass the “Comments to the Author” section, enter your conflict of interest statement in the “Confidential to Editor” section, and submit your "Accept" recommendation.

Reviewer #1: All comments have been addressed

Reviewer #2: All comments have been addressed

Reviewer #3: All comments have been addressed

2. Is the manuscript technically sound, and do the data support the conclusions?

Reviewer #1: Yes

Reviewer #2: (No Response)

Reviewer #3: Yes

3. Has the statistical analysis been performed appropriately and rigorously? 

Reviewer #1: Yes

Reviewer #2: (No Response)

Reviewer #3: Yes

4. Have the authors made all data underlying the findings in their manuscript fully available?

Reviewer #1: Yes

Reviewer #2: (No Response)

Reviewer #3: Yes

5. Is the manuscript presented in an intelligible fashion and written in standard English?

Reviewer #1: Yes

Reviewer #2: (No Response)

Reviewer #3: No

6. Review Comments to the Author

Reviewer #1: Thank you for addressing comments from reviewers so thoroughly. I have only a few comments / clarifications.

I suggest you reword the following updated sentence (in the Statistical analysis, page 7, paragraph 1). "To possibly obtain sufficient statistical power...." replace that with "To increase statistical power". This might just be poor English, but it's not correct.

In the discussion section, (page 19, paragraph 2), a sentence reads "First, the small sample size in our study may led to a lack of statistical significance". This, too, may be poor English expression (I note that the original manuscript has been edited by an native English speaker) but this is not correct either. I presume the authors mean that the number of events observed was low, leading to inaccurate estimates of effect size.

Reviewer #2: (No Response)

Reviewer #3: (No Response)

7. PLOS authors have the option to publish the peer review history of their article (what does this mean?). If published, this will include your full peer review and any attached files.

Reviewer #1: No

Reviewer #2: No

Reviewer #3: No

---

## [Author Response · Author response to Decision Letter 1]

10 Aug 2020

Reviewer #1: Thank you for addressing comments from reviewers so thoroughly. I have only a few comments / clarifications.

1. I suggest you reword the following updated sentence (in the Statistical analysis, page 7, paragraph 1). "To possibly obtain sufficient statistical power...." replace that with "To increase statistical power". This might just be poor English, but it's not correct.

Ans: Thank you for pointing out the correct description. We have corrected this error in the new version.

On line 12-14, page 7: 

“To increase statistical power, PS matching pair from 1:1 to 1:5 using the Greedy 5 to 1 digit technique [15] was taken into consideration.” 

2. In the discussion section, (page 19, paragraph 2), a sentence reads "First, the small sample size in our study may led to a lack of statistical significance". This, too, may be poor English expression (I note that the original manuscript has been edited by an native English speaker) but this is not correct either. I presume the authors mean that the number of events observed was low, leading to inaccurate estimates of effect size.

Ans: Thank you for the valuable suggestion. We have made additional English editing by another native English speaker. This error has been corrected in the new version. 

On line 30-32, page 30:

“Firstly, the sample size was small, though it was representative, and the information in Taiwan’s dialysis database is relatively more complete than that found in other countries.”

---

## [Editor Report · Decision Letter 2]

13 Aug 2020

Effectiveness of antiresorptive medications in women on long-term dialysis after hip fracture: A population-based cohort study

PONE-D-20-07988R2

Dear Dr. Lin,

We’re pleased to inform you that your manuscript has been judged scientifically suitable for publication and will be formally accepted for publication once it meets all outstanding technical requirements.

Kind regards,

Yuanyuan Wang

Academic Editor

PLOS ONE

Additional Editor Comments (optional):

The authors have addressed all the reviewers' comments.
---

## [Editor Report · Acceptance letter]

18 Aug 2020

PONE-D-20-07988R2 

Effectiveness of antiresorptive medications in women on long-term dialysis after hip fracture: A population-based cohort study 

Dear Dr. Lin:

I'm pleased to inform you that your manuscript has been deemed suitable for publication in PLOS ONE. Congratulations! Your manuscript is now with our production department. 

Kind regards, 

on behalf of

Dr. Yuanyuan Wang 

Academic Editor

PLOS ONE